# Molecular-Simulation–Inspired Synthesis of [6]-Prismane via Photoisomerisation of Octafluoro[2.2]paracyclophane

**DOI:** 10.3390/molecules29040783

**Published:** 2024-02-08

**Authors:** Yoichi Hosokawa, Shuji Kajiya, Ayako Ohshima, Satoshi Kawata, Nobuhiro Ishida, Arimitsu Usuki

**Affiliations:** 1Toyota Central R&D Labs., Inc., 41-1, Yokomichi, Nagakute, Aichi 480-1192, Japan; e1181@mosk.tytlabs.co.jp (S.K.); aya-o@mosk.tytlabs.co.jp (A.O.); n-ishida@mosk.tytlabs.co.jp (N.I.); usukifam20@gmail.com (A.U.); 2Department of Chemistry, Faculty of Science, Fukuoka University, 19-1 Nanakuma 8-Chome, Jonan-ku, Fukuoka 814-0180, Japan; kawata@fukuoka-u.ac.jp

**Keywords:** prismane, fluorine group, cyclophane, photoisomerisation, molecular mechanics, mopac, density functional theory, frontier orbital theory

## Abstract

Prismanes have been attracting interest for nearly 50 years because of their geometric symmetry, highly strained structures, and unique applications due to their high carbon densities and bulky structures. Although [3]-, [4]-, and [5]-prismanes have been synthesised, [6]-prismanes and their derivatives remain elusive. Herein, fluorine chemistry, molecular mechanics, molecular orbital package, and density functional theory calculations were used to design and implement the photoisomerisation of octafluoro[2.2]paracyclophane (selected based on the good overlap of its lowest unoccupied molecular orbitals and short distance between the benzene rings) into octafluoro-[6]-prismane. Specifically, a dilute solution of the above precursor in CH_3_CN/H_2_O/dimethyl sulfoxide (DMSO) (2:1:8, *v*/*v*/*v*) solution was irradiated with ultraviolet light, with the formation of the desired product confirmed through the use of nuclear magnetic resonance spectroscopy and gas chromatography–mass spectrometry. The product was thermally stable in solution but not under work-up conditions, which complicated the further analysis and single-crystal preparation. The key criteria for successful photoisomerisation were the presence of fluorine substituents in the cyclophane structure and DMSO in the solvent system. A more stable derivative design requires the isolation of prismane products. The proposed fluorination-based synthetic strategy is applicable to developing novel high-strain molecules/materials with three-dimensional skeletons.

## 1. Introduction

Prismanes, a class of hydrocarbons with prism-like structures featuring interconnected cyclobutane sides attached to polygonal bases, exhibit high strain energies owing to their 90° C–C bond angles (Figure 1) [1,2]. The syntheses of [3]- (1), [4]- (2), and [5]-prismanes (**3**), confirmed through the use of nuclear magnetic resonance (NMR) spectroscopy and mass spectrometry (MS), were first reported in 1973, 1964, and 1981, respectively [3,4,5], with the structures of the [3]- and [4]-prismane skeletons determined via single-crystal X-ray diffraction in 1987 and 1964, respectively [6,7]. [4]-Prismanes (cubanes) have found medicinal applications and hold promise for the fabrication of high-energy materials and high-strength fibres because of their chemical and thermal stabilities [8].

In the 40 years following the synthesis of **3**, considerable efforts have been made to synthesise [6]-prismane (**5**). Despite the success of the photocatalytic stepwise synthesis of secohexaprismane (**4**) from a Diels–Alder adduct [9,10], **5** and its derivatives remain unknown, largely because of their high strain energy (682 kJ/mol for **5** vs. 569, 632, and 540 kJ/mol for **1**, **2**, and **3**, respectively) [11]. Diamond nanothreads (DNTs, **6**) are a new class of hydrocarbon polymers with a three-dimensional (3D) structure similar to that of **5**. These nanothreads, produced by the isomerisation of benzene at pressures below 20 GPa, are predicted to be stronger (stiffness = 850 GPa) than carbon nanotubes (2 GPa), making them the strongest material currently available for constructing space-elevator cables (required specification: 50 GPa) [12,13,14]. The synthesis of [6]-prismanes should also be useful for developing higher prismanes and other molecules/materials with 3D skeletons, such as capsule-form polycyclic aromatic hydrocarbons/coronenes [15,16,17,18].

[6]-Prismanes can potentially be synthesised via the photoreactions of cyclophanes, which are macrocycles featuring two benzene rings bridged by an aliphatic chain (Figure 2). Although the photoreaction of [2.2]cyclophane (**7**) results in macrocycle decomposition only (Figure 2a) [19], that of [2.2] naphthalenophane (**8**) affords a product cross-linked at different positions (Figure 2b) [20]. The photoreaction of [3.3]cyclophane (**9**, Figure 2c) [21] involves the formation of C–C bonds between its benzene rings through acidic protons and the reaction of these rings with the solvent, i.e., water or methanol, yielding OH- or methoxy-substituted compounds, respectively. This behaviour is ascribed to the low reactivity of the benzene rings and flexibility of the long alkyl chain linkers. The above results suggest that the photoreactivity of the two benzene rings can be controlled by adjusting the electronic and steric properties of the functional groups, such as the number and position of the methylene units and electron-donating/-withdrawing ability.

Previously, theoretical calculations have been conducted to examine the structural and thermodynamic properties of prismanes and extended compounds, such as prismane polymers and silicon prismanes, and probe the effect of substitution on their mechanical properties. However, numerous known (e.g., DNTs) and unknown compounds are yet to be studied [22,23,24,25,26,27]. Herein, molecular mechanics (MMs), the molecular orbital package (MOPAC) PM7 software (version 16.093W 64bits), and density functional theory (DFT) calculations were used to design the synthesis of a novel fluorinated [6]-prismane derivative (**11**) based on one-pot photoisomerisation, namely the [6+6] intramolecular cycloaddition of the commercially available octafluoro[2.2]paracyclophane (**10**, Figure 2d and Appendix A). The product was thermally stable in the reaction solution but unstable in alcohols and during work-up, which complicated isolation and the growth of single crystals for X-ray diffraction analysis. The key criteria for the success of the photoisomerisation were identified as the introduction of fluorine atoms and the use of dimethyl sulfoxide (DMSO) in the solvent system. The developed simulation-based strategy relying on fluorination is expected to be applicable to the prediction and synthesis of novel high-strain molecules and materials.

## 2. Results

### 2.1. Reaction Design

First of all, the effects of substituents on the steric stability of the prismane skeleton were evaluated using MM2 calculations (Appendix A). This skeleton was retained in the case of halogen atoms and methylene bridges [(CH_2_)_2_], whereas other substituents, such as alkyl, alkoxy, and cyano groups, caused a transition to the boat or chair form because of steric repulsion. Among the various halogenated compounds, fluorinated ones are generally stable; therefore, fluorinated [2.2]cyclophanes were selected as prospective starting material candidates for [6]-prismane synthesis. The electron-withdrawing fluorine groups were expected to increase the reactivity of the benzene rings, the distance between them, and the stability of the [6]-prismane structure, as similar structural stabilisation has been predicted for a fluorinated Si-[6]-prismane [27].

Next, MM2 structures of the selected precursors (**10**, **12**, and **14**) and their cyclisation products (**11**, **13**, and **15**, respectively) were optimised using the MOPAC PM7 software (see Appendix A for structures). The heats of the formation (kcal/mol) of these compounds (Appendix A) showed that cyclisation was endothermic in all cases (by approximately 30, 40, and 120 kcal/mol for **12** → **13**, **14** → **15**, and **10** → **11**, respectively). The conversion of **10** to **11** was predicted to result in a C···C distance shortening of 1.2–1.6 Å (Figure 3c and Appendix A).

Furthermore, the distances between the benzene rings (hereinafter referred to as ring distances) and the overlap between the lowest unoccupied molecular orbitals (LUMOs) were investigated using DFT calculations. More specifically, the intramolecular formation of C–C bonds between the benzene rings was focused. Typically, **12** is converted to **13** via a [2+2] intramolecular photocycloaddition [28,29]. The distance between the two double bonds of **12** is very short (~2.4 Å), and the LUMOs localised on these bonds exhibit good overlap (Figure 3a).

For the boat-type cyclophane structure, the two distances correspond to cross-linked and non-cross-linked positions. The ring distance in fluorinated [3.3]cyclophane **16** was longer than that in [2.2]cyclophanes **7** and **10**. Among those in [2.2]cyclophanes, the ring distance in cyclophane **10** was somewhat shorter than that in the parent cyclophane (**7**) (Figure 3b–d).

Although the LUMOs of cyclophanes **10** and **16** were localised at the cross-link carbons, this was not the case for the parent cyclophane (**7**). Furthermore, cyclophane **10** exhibited a larger orbital overlap than **7** and **16**, as expected from its shorter ring distance. The largest and smallest occupied molecular orbital HOMO–LUMO gaps were observed for **12** and **10**, respectively (Appendix A). Hence, **10** was selected as the starting material for photocyclisation.

### 2.2. Photoreaction of ***10***

The photoreaction of **10** in CD_3_CN/D_2_O/(CD_3_)_2_SO (2:1:8, *v*/*v*/*v*; 6.25 mM solution) was performed under ultraviolet (UV) irradiation (240–400 nm, 300 W Xe lamp) in a quartz NMR tube. The reaction was monitored using NMR spectroscopy, and the reaction solution before and after the reaction was analysed via gas chromatography–mass spectrometry (GC–MS). In a scale-up test, the reaction was performed at an approximately six-fold larger scale using an irradiation time of 4 h.

Figure 4 shows the time-dependent ^19^F NMR spectra recorded during the photoreaction of **10**, revealing the emergence of a new peak corresponding to **11** at approximately −170 ppm after 15 min (Figure 4a). After 45 min, this peak was the only signal observed, and that of **10** at approximately −140 ppm disappeared (Figure 4b). The chemical shift of the former peak (−170 ppm) is between those of *exo*- (−165.5 ppm) and *endo*-fluorocyclohexane (−186.0 ppm) and is, therefore, reasonable [30]. Similarly, the methylene proton peak at ~3.25 ppm disappeared after UV irradiation (Appendix A).

UV–Vis spectroscopy could not be used for reaction monitoring because UV irradiation induced a side reaction even in the solvent-only system (Appendix A, red line). The irradiated reaction solution was pale yellow (Appendix A, purple line), which, again, complicated the use of UV–Vis spectroscopy as a diagnostic tool.

A low cyclophane concentration (2 mg/mL) was used to prevent intermolecular reactions such as polymerisation. The reaction was successfully performed without the formation of insoluble precipitates in CH_3_CN/H_2_O/ DMSO (2:1:8, *v*/*v*/*v*). These three solvents were selected to achieve starting material dissolution (CH_3_CN), product dissolution (DMSO), and reaction acceleration (H_2_O). When CH_3_CN, DMSO, or a CH_3_CN/DMSO mixture was used, almost no reaction was observed, and **10** was insoluble in H_2_O or H_2_O/CH_3_CN. Therefore, a mixed solvent, such as H_2_O/DMSO or CH_3_CN/H_2_O/DMSO, was essential for the photoreaction of **10** (Appendix A). The ^1^H NMR spectra recorded using the D_2_O/DMSO-*d*_6_ system mentioned in Appendix A revealed that the CH_2_ peak shifted from 3.18 ppm to 1.98 ppm (Appendix A) after UV irradiation.

The product (**11**) was thermally stable in the reaction solution (Appendix A), as revealed by the persistence of its ^19^F signal (−170 ppm) after the reaction mixture was heated at 80–100 °C for 5 h in the NMR tube, which also suggested that the reaction was driven by UV irradiation rather than heat.

### 2.3. Isolation and Characterisation of ***11***

Given that **4** is a highly volatile waxy solid that sublimes at 80 °C (mp > 250 °C) [9], **11** was also expected to be volatile and prone to sublimation because of its highly fluorinated structure. All tested product isolation and single-crystal preparation methods were ineffective owing to the low reaction concentration (2 mg/mL) and the presence of DMSO as a major solvent. No residue was obtained after solvent removal via vacuum distillation and freeze-drying. Solvent extractions using CH_2_Cl_2_, CHCl_3_, ethers, and hydrofluoroethers were also unsuccessful. Purification via column chromatography (e.g., silica, alumina, and preparative gel permeation chromatography) caused decomposition. Furthermore, the use of alcohols caused reactions/decompositions, as reported previously [21].

Figure 5 shows the GC–MS analysis results of pristine and irradiated samples. After UV irradiation, the peak of the starting material at ~17 min (Figure 5a) disappeared, and a new peak emerged at 16 min (Figure 5b). The molecular weights corresponding to these two peaks were the same (*m/z* = 352), indicating that the observed transformation corresponded to photoisomerisation. Considering the high symmetry indicated by the results of NMR analysis, these observations agreed with the formation of **11**.

The ^13^C NMR spectrum of the reaction solution (100 MHz, ~70,000 scans) showed only the solvent peak because of the low solution concentration (~1.2 mg/0.6 mL, CD_3_CN/D_2_O/DMSO-*d*_6_ = 2/1/8, *v*/*v*/*v*), as shown in Appendix A. Consequently, we explored high-concentration reaction conditions using different solvent ratios and found that an approximately 4.6-fold higher cyclophane concentration could be obtained using a high acetonitrile content (i.e., 8.8 mg/0.96 mL, CD_3_CN/D_2_O/DMSO-*d*_6_ = 8/1/2, *v*/*v*/*v*). However, the photoreaction was unsuccessful because of the formation of an insoluble pale-yellow material (possibly impurities in **10** and decomposedpolymer/aggregation products) on the inner surface of the NMR tube, which blocked UV light.

Subsequently, acetonitrile was removed from the reaction mixture by evaporation to obtain a highly concentrated solution. The photoreaction of the original low-concentration solution with a high acetonitrile content (i.e., 1.2 mg/0.6 mL, CD_3_CN/D_2_O/DMSO-*d*_6_ = 8/1/2, *v*/*v*/*v*) was successful, and an approximately 4.0-fold higher concentration (i.e., 4.8 mg/0.6 mL) was obtained for ^13^C NMR measurements via direct solvent evaporation under vacuum for several minutes. The sp^2^ carbon peaks of the starting material (**10**) at ~118 and 146 ppm (Appendix A) disappeared after UV irradiation and were replaced by sp^3^ carbon signals at ~17, 22, and 82 ppm (Appendix A) and a septet-like peak at ~42 ppm, probably due to solvent derived impurity. Considering the ^13^C NMR data of fluorocyclohexane (~24/t, 26/t, 34/t, and 92/d ppm in CDCl_3_) [31], the peaks at ~17, 22, and 82 ppm were ascribed to methylene-bridge, and strained-position fluorinated sp^3^ carbons, respectively. Additionally, acetonitrile removal was attempted to clearly observe the sp^3^ carbon peak. However, long-term vacuum treatment (~10 min) yielded a highly viscous solution affording a broad peak in the ^19^F NMR spectrum, which complicated further investigation (Appendix A).

On the other hand, the combination of a low-concentration photoreaction solution also yielded a similar result as shown in Appendix A. Therefore, we asked JEOL to perform ^1^H/^19^F -decoupled ^13^C NMR and two-dimensional NMR measurements using the special multi-decoupling probe (ROYAL probe HFX with JEOL JNN-ECZ600) to obtain clear signals. The non-irradiated solution of cyclophane **10** showed the signals clearly (Appendix A). Unfortunately, the irradiated solution of **10** formed impurities during 0.5–1 day low-temperature shipping from TCRDL to JEOL. Although no valid signals were observed in the two-dimensional NMR measurement, three decoupled peaks as potential candidates were observed at 36.5, 46.5, and 81.5 ppm with some impurity peaks in the ^13^C {^1^H, ^19^F} spectra (Appendix A, carbon label a–c)

The crystal structure of **10** was determined using a CH_3_CN–MeOH vapour system (Appendix A). No suitable solvent or preparation method was found for the synthesis of single-crystalline **11** because of the difficulty of purification/isolation caused by the low concentration of the DMSO-containing reaction mixture, as described above. However, we recovered trace amounts of the precipitate from the reaction solution by adding a 10-fold volume of water for single-crystal formation. The corresponding ATR-IR spectrum (Appendix A) featured no sp^2^ carbon peak of **10** but contained the sp^3^ carbon peak expected for **11**.

## 3. Discussion

In the case of **8**, OH substitution on the benzene ring and the addition of 1 N HCl to a CH_2_Cl_2_/H_2_O mixed solvent were reported to accelerate the reaction [17], i.e., the benzene ring was activated by protonation. Thus, the reaction was suggested to involve the formation of a carbocation and its intramolecular rearrangement. Although the reaction occurred in a mixed aqueous solvent, a fluorinated compound was used as the starting material. Thus, the presence of H_2_O in the reaction solution was expected not to cause OH substitution but rather stabilise the fluorinated product to accelerate isomerisation due to hydrophobic interactions. In addition, the crowded fluorine atom surface in **11** was expected to prevent the attack of OH^−^ in the presence of H_2_O. Considering the immediate product formation, simple NMR spectral changes and chemical shifts, high thermal stability of the product, and GC–MS results, the formation of **11** was concluded to proceed via a [6+6] intramolecular cycloaddition.

Alternatively, we attempted solid-phase photoisomerisation for a milled single crystal of **10** to obtain the molecular structure of **11**. For a solid-state photoreaction to occur, the distance between the double bonds should be <4.2 Å (Schmidt’s rule) [32,33]. In our system, the distance between the two benzene rings was 3.0 Å, satisfying this condition (Appendix A, front view). However, the reaction was unsuccessful and resulted in the recovery of the starting material, as confirmed through the use of NMR spectroscopy. This failure was attributed to the rigidity of the cyclophane structure in the solid state. As predicted by the molecular simulation of **12**, we suggested that a distance <2.5 Å is required for intramolecular bond formation to take place in the rigid molecule.

The fluorine groups were used to activate the benzene rings. Octanitrocubane, which has a cubic structure with eight NO_2_ groups, is a well-known high-energy explosive [34]. Although the nitro group may increase the cyclophane reactivity, side reactions, such as N–O and C–N bond cleavage could occur under the employed conditions. Low solubility was also expected for fully NO_2_-substituted prismanes.

Inokuma et al. developed crystal sponge and liquid chromatography–single crystal diffraction methods for crystal structure analyses [35,36]. Matsumoto et al. reported a molecular grabber method that does not require the direct crystallisation of the target molecules [37]. Although the availability of our DMSO-containing solution for the above methods is not known, approaches using the absorption/inclusion phenomena for Metal–Organic Frameworks (MOFs) and proteins are expected to be suitable for the structural analysis of this unstable product.

Multibridged cyclophanes have attracted attention as supramolecular building blocks, such as electron donors [38] and cage compounds for noble gas atoms [39]. Various cyclophane derivatives with different ring distances have been reported. In particular, [2.6]cyclophane has a ring distance of ~2.65 Å. Multibridged fluorocyclophanes, featuring some of the shortest ring distances, can potentially form stable single crystals. Furthermore, strained fluorinated molecules, such as perfluorocubane (perfluoro-[4]-prismane) [40] and perfluorocycloparaphenylenes [41], have been reported as stable crystals. The ^19^F NMR shift of the fluorine atom in perfluoro-[4]-prismane is 197.19 ppm (475 MHz, acetone-*d*_6_), and the ^13^C NMR shift of the fluorinated carbon is 103.78 ppm (125 MHz, acetone-*d*_6_), which is attributed to the highly strained perfluorinated structure. The introduction of fluorine atoms is probably effective for the stabilisation of strained molecules, and perfluoro-[6]-prismane should, therefore, undergo stable crystallisation.

## 4. Materials and Methods

### 4.1. Materials and Measurements

All reagents were used as received without further purification. Octafluoro[2.2]paracyclophane (**10**, >98%) was purchased from Tokyo Chemical Industry (Tokyo, Japan). Acetonitrile (99.5%), ultrapure water, and DMSO (99.0%) were obtained from FUJIFILM Wako Pure Chemical Corporation (Osaka, Japan).

UV irradiation was carried out using an Asahi Spectra (Tokyo, Japan) MAX-303 Xe lamp (300 W, 240–400 nm). The UV spectra were obtained using TECAN Spark-Fusion (Tecan Group Ltd., Männedorf, Switzerland).

NMR spectra were acquired using a JEOL (Tokyo, Japan) JMM-EX400 spectrometer. Decoupling and two-dimensional NMR spectra were measured using a JEOL JNN-ECZ600 spectrometer with ROYAL(Reverse Optimally Autotune Liquid) probe HFX.

GC–MS measurements were performed using an Agilent GC/MS 7890B/5977B instrument (Agilent Technologies, Santa Clara, CA, USA) under the following conditions: sample volume, 1.0 μL; column, Agilent DB5-MS; oven temperature, 40 °C (hold time: 1 min), increased at 10 °C/min to 280 °C (3 min); injection temperature, 280 °C; elution speed, 1.5 mL/min; split ratio, 10:1; carrier gas, He; transfer line temperature, 280 °C; ionisation method, electrospray ionisation; MS ion source temperature, 230 °C.

Attenuated total reflection–Fourier transform infrared (ATR-FTIR) spectra were acquired on a ThermoFisher Scientific (Waltham, MA, USA) Nicolet Avatar 360 FT-IR spectrometer (scan: 60, resolution: 8).

Single-crystal X-ray diffraction analysis was performed using a Rigaku Corporation R-AXIS RAPID instrument (Tokyo, Japan). All structural calculations were performed using the Crystal Structure crystallographic software package Version 4.2.2., except for refinement, which was performed using SHELXL Version 2016/6.

### 4.2. General Procedure for the Photoreaction of ***10***

Cyclophane **10** (8.0 mg, 6.25 mM) was dissolved in CD_3_CN/D_2_O/(CD_3_)_2_SO (2:1:8, *v*/*v*/*v*; 4 mL). The resulting solution (0.6 mL) was placed in a quartz NMR tube and subjected to UV irradiation (240–400 nm). The scaled-up experiment was performed in a 10 mL quartz test tube (irradiated volume = 3.75 mL) using the same solvent mixture and reaction concentration: ^19^F NMR (373 MHz, DMSO-*d*_6_, CFCl_3_ standard, Appendix A) for **10**: δ −140.55; **11**: δ −169.69. ^13^C NMR (100 MHz, CD_3_CN/D_2_O/(CD_3_)_2_SO) for **10**: δ 20.82, 117.76, 146.29 (d, ^1^*J*_C-F_ = 251.4 Hz); **11**: δ 36.5 46.5, 81.5. ^1^H NMR (400 MHz, DMSO-*d*_6_) for **11**: δ 1.98.

### 4.3. Calculation Environments

All programs were run on a Dell Precision T7910 PC (CPU: Xeon E5-2630 v3 (2.40 GHz), RAM: 32 GB, OS: Windows 7 Pro SP1). ChemDraw Professional 15.1 was used to draw chemical structures. The Chem3D 15.1 MM2 calculation (minimise energy program) was used for initial structural calculations, and the Chem3D 15.1 MOPAC 2016 program was used for structure optimisation and heat of formation calculations [42]. The Materials Studio 7.0 SP2 DMol3 module was used for HOMO–LUMO calculations [43]. All calculations were performed on a remote server (CPU: Xeon E5-2690 × 2 (SandyBridge/8 core/2.90 GHz), RAM: 64 GB, InfiniBand: FDR 56 Gbps, OS: RHEL6.1).

### 4.4. General Procedure for Optimising Functionalised [6]-Prismane Derivatives Using MM2 Calculations

A [6]-prismane structure drawn in ChemDraw was pasted into Chem3D for MM2 optimisation (minimise energy program) for 2–3 min to remove any initial structural overlap, and each functional group was then pasted into Chem3D and connected to the [6]-prismane structure. Subsequently, MM2 calculations were repeated.

### 4.5. General Procedure for MOPAC PM7 Calculations

MOPAC PM7 calculations were performed on each MM2-optimised structure using the following parameters: job type, minimisation (energy/geometry); method, PM7; properties, heat of formation; all other parameters, default settings. Appendix A shows the Cartesian coordinates of the PM7 optimized geometry.

### 4.6. General Procedure for HOMO–LUMO Calculations

MS DMol3 calculations were performed on each PM7-optimised structure using the following parameters: task, energy; quality, medium; functional, GGA-BLYP; smearing, 0.005 Ha; basis, DND; properties, Fukui function, orbitals, and population analysis; all other parameters, default settings.

## 5. Conclusions

A photoisomerisation reaction of octafluoro[2.2]paracyclophane (**10**) was designed to synthesise a [6]-prismane derivative based on MM, MOPAC, and DFT calculations. The formation of octafluoro-[6]-prismane following the UV irradiation (240–400 nm) of **10** in CH_3_CN/H_2_O/DMSO (2:1:8, *v*/*v*/*v*) was confirmed through the use of UV/Vis, IR, NMR, and GC–MS analyses based on previous reports on [3]-, [4]-, and [5]-prismanes. Although thermally stable in the reaction solution, the product was unstable in alcohols and under various work-up conditions; as a result, isolation/single-crystal preparation approaches were ineffective. The key criteria for the photochemical reaction were the introduction of fluorine atoms and the use of DMSO in the solvent system. The present simulation-based strategy relying on fluorination is expected to be applicable to developing novel high-strain molecules and materials. Further efforts and approaches towards structural analysis, as well as investigations of other stable derivatives, are in progress, with a report on the [6]-prismane skeleton expected in the near future.

## Figures and Tables

**Figure 1 molecules-29-00783-f001:**
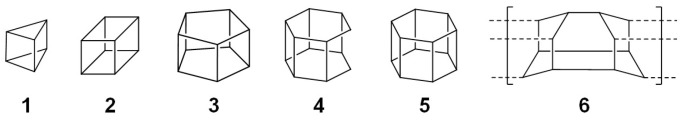
Prismanes and related compounds.

**Figure 2 molecules-29-00783-f002:**
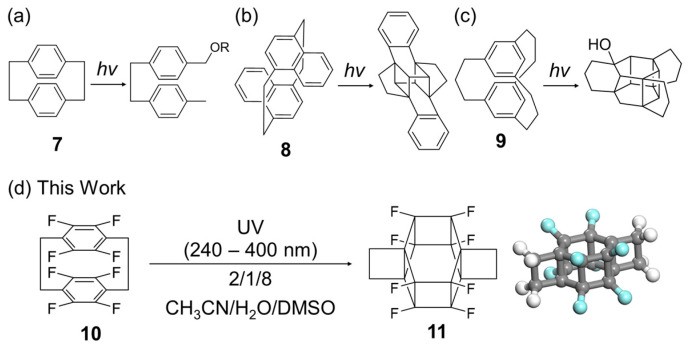
Photoreactions of (**a**) [2.2]cyclophane (**7**), (**b**) [2.2]naphthalenophane (**8**), (**c**) [3.3]cyclophane (**9**), and (**d**) octafluoro[2.2]paracyclophane (**10**). The chemical structure and 3D model of [6]-prismane (**11**). Carbon atom: gray, fluorine atom: pale blue, hydrogen atom: white.

**Figure 3 molecules-29-00783-f003:**
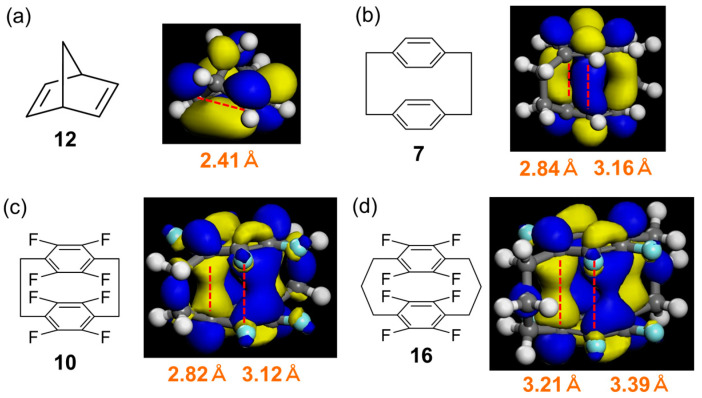
Optimised structures, lowest unoccupied molecular orbitals (LUMOs), and ring distances of (**a**) norbornadiene (**12**), (**b**) [2.2]cyclophane (**7**), (**c**) fluorinated [2.2]cyclophane (**10**), and (**d**) fluorinated [3.2]cyclophane (**16**). Dotted red lines represent C···C distances. Carbon atom: gray, fluorine atom: pale blue, hydrogen atom: white.

**Figure 4 molecules-29-00783-f004:**
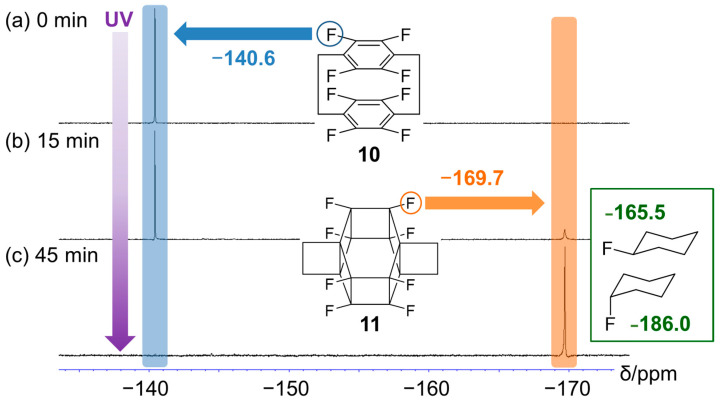
^19^F nuclear magnetic resonance (NMR) spectra (400 MHz, CD_3_CN/D_2_O/dimethyl sulfoxide (DMSO)-*d*_6_ = 2:1:8, *v*/*v*/*v*) recorded after the photoreaction of **10** for (**a**) 0, (**b**) 15, and (**c**) 45 min. Blue area: sp^2^ carbon fluorine, orange area: sp^3^ carbon fluorine.

**Figure 5 molecules-29-00783-f005:**
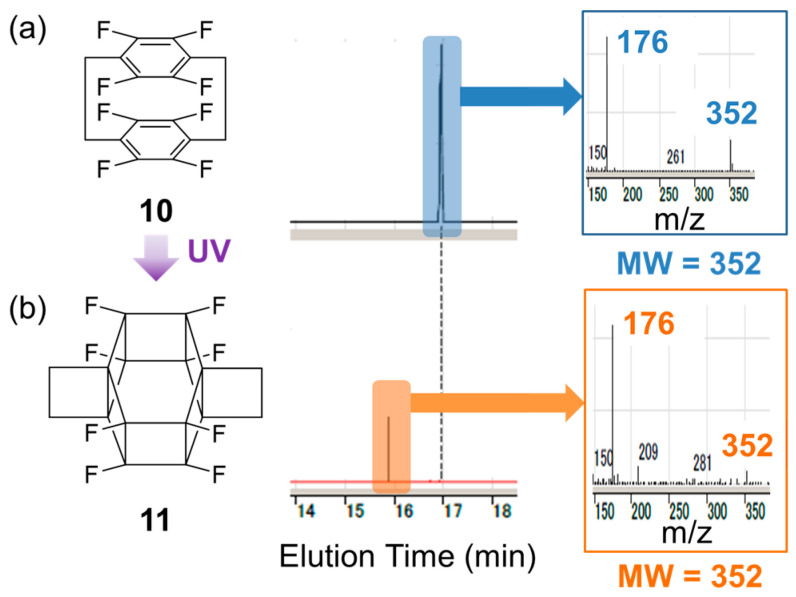
Gas chromatography–mass spectrometry analysis of the reaction solution (**a**) before and (**b**) after irradiation.

## Data Availability

The X-ray crystallographic coordinates of the structure reported in this study were deposited at the Cambridge Crystallographic Data Centre (CCDC) under the deposition number CCDC-2003703 (compound **10**). These data can be obtained free of charge from the CCDC at www.ccdc.cam.ac.uk/data_request/cif (accessed on 2 February 2024). All other data supporting the findings of this study are available from the corresponding author upon request.

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
