# Peer review of "Molecular-Simulation–Inspired Synthesis of [6]-Prismane via Photoisomerisation of Octafluoro[2.2]paracyclophane"

_molecules, 2024, doi:10.3390/molecules29040783_

Round 1
Reviewer 1 Report
Comments and Suggestions for Authors
In the manuscript, photoisomerization of octafluorinated paracyclophane was studied. The resulting prismane structure was identified by NMR spectroscopy in solution. The starting compound was preliminary evaluated by quantum chemical calculations. In terms of experiment, which is the most important part, the paper is nice. However, in terms of quantum chemistry, there are some concerns. The major point is that why the authors used force field calculations for relatively small systems? Nowadays, even regular PCs can handle these systems by DFT (full optimization may be done by some DFT method). Overall, the manuscript may be accepted after minor revision taking into account comments given below.
Comments:
1. It would be interesting to compare the ground state energies of starting 10 and product 11. This may be calculated by DFT methods, and the energies may be provided (complete optimization is needed).
2. For quantum chemical calculations, it is a routine to validate the stationary points by frequency calculations. Was this done?
3. Usually, for reporting calculation results, final coordinates and energies are provided in Supporting Information.
4. Which basis set for DFT method was used? This should be unequivocally provided.
5. The structure of prismane 11 also may optimized. What is the decrease of C…C bond on shifting from starting 10 and product 11?
Reviewer 2 Report
Comments and Suggestions for Authors
In their publication “Molecular Simulation, Reaction Design, and Photoisomerization of Octafluoro[2.2]paracyclophane for [6]-Prismanes” Hosokawa et. al. present the quantumchemical design and photochemical synthesis of fluorinated [6]-prismanes. Although the data shown are presented clearly, additional experiments and clarifications are required before publishing this work:
1. I am surprised that the authors did not record UV-Vis spectra of product and educt. This could be a very easy way to observe the conversion after UV irradiation, especially since at least the educt should have absorption between 250 and 400 nm, considering the irradiation source.
2. The authors showed 13C NMR spectra after irradiation with a very low signal-to-noise ratio. This was attributed to the low concentration that must be used to avoid the formation of an insoluble material. I am not convinced that the signals assigned in Figure S6c (especially the carbon atoms labelled “a” and “b”) are really signals and not just random noise. In addition, there appears to be a signal at approximately 42 ppm, which has not been discussed at all. Therefore, it would be beneficial to repeat this experiment. A possible way would be to combine different batches of low-concentration solutions after irradiation to obtain a better signal-to-noise ratio while avoiding the formation of the insoluble product.
3. Can the authors clarify what the insoluble product is, which is formed upon irradiation with higher-concentration solutions?
4. In 1H-NMR, the authors clearly showed the disappearance of the educt signal upon irradiation. However, the formation of the product signal cannot be detected because it is superimposed by the acetonitrile signal. The experiment could be repeated in D2O/(CD3)2SO to clearly show product formation, as the authors showed that the product was soluble in this mixture.
5. On page 4, line 133, and in Figure 4, it is mentioned that in 19F NMR after 15 min of irradiation, a signal corresponding to cyclophane is detected. So, what does the spectrum of the parent solution look like after 0 min of irradiation?
After these revisions and clarifications have been made, the manuscript will be suitable for publication in Molecules.
Comments on the Quality of English Language
There were a few grammatical and spelling errors in the manuscript.
Therefore, I advise the authors to carefully proofread the manuscript after incorporating their corrections.
Round 2
Reviewer 2 Report
Comments and Suggestions for Authors
The authors significantly improved their manuscript now titled “Molecular-Simulation–Inspired Synthesis of [6]-Prismane via Photoisomerisation of Octafluoro[2.2]paracyclophane” by performing additional experiments, namely UV-Vis, NMR and IR measurement. These experiments answered the questions which arose from the first version of the manuscript. Therefore, the manuscript in its current version is suitable for publication in Molecules.